# How medical education survives and evolves during COVID-19: Our experience and future direction

Ju Whi Kim[1], Sun Jung Myung[1]*, Hyun Bae Yoon[1], Sang Hui Moon[1], Hyunjin Ryu[1], Jae-Joon Yim[1,2]

1 Office of Medical Education, Seoul National University College of Medicine, Seoul, South Korea,
2 Department of Internal Medicine, Seoul National University College of Medicine, Seoul, South Korea

* issac73@snu.ac.kr

## Abstract

### Background

Due to the outbreak of coronavirus disease 2019 (COVID-19), school openings were postponed worldwide as a way to stop its spread. Most classes are moving online, and this includes medical school classes. The authors present their experience of running such online classes with offline clinical clerkship under pandemic conditions, and also present data on student satisfaction, academic performance, and preference.

### Methods

The medical school changed every first-year to fourth-year course to an online format except the clinical clerkship, clinical skills training, and basic laboratory classes such as anatomy lab sessions. Online courses were pre-recorded video lectures or live-streamed using video communication software. At the end of each course, students and professors were asked to report their satisfaction with the online course and comment on it. The authors also compared students' academic performance before and after the introduction of online courses.

### Results

A total of 69.7% (318/456) of students and 35.2% (44/125) of professors answered the questionnaire. Students were generally satisfied with the online course and 62.2% of them preferred the online course to the offline course. The majority (84.3%) of the students wanted to maintain the online course after the end of COVID-19. In contrast, just 13.6% of professors preferred online lectures and half (52.3%) wanted to go back to the offline course. With the introduction of online classes, students' academic achievement did not change significantly in four subjects, but decreased in two subjects.

### Conclusions

The inevitable transformation of medical education caused by COVID-19 is still ongoing. As the safety of students and the training of competent physicians are the

(contact via Tel: 82-2-2072-0694) for researchers who meet the criteria for access to confidential data.

**Funding:** The authors received no specific funding for this work.

**Competing interests:** The authors have declared that no competing interests exist.

responsibilities of medical schools, further research into how future physicians will be educated is needed.

## Introduction

Medical education has gradually been changing and one significant part of this has been the introduction of online learning, which is now widespread not only in medical education but in many other fields [1]. Online learning has been demonstrated to be as effective as conventional didactic teaching and can be used to promote self-directed learning [2, 3]. According to a recently published meta-analysis, blended learning for the medical professions comprising face-to-face learning and online learning has increased knowledge compared with education using only one or the other method [4]. However, many medical schools in Korea are still sticking to face-to-face lectures and many professors prefer offline lectures rather than online ones.

Schools are closed in many parts of the world to alleviate the outbreak of COVID-19 [5]. In the Republic of Korea, with the exponential increase in the number of confirmed cases when a number of cases of regional infections involving Daegu and Gyeongbuk area related to religious gatherings (Shincheonji) have been reported, the Ministry of Education postponed the start of the new school year until late May 2020 [6]. Moreover, the risk alert level for infectious diseases has been upgraded to "serious." We no longer have the opportunity to choose between online and offline lectures. The time has come to move all face-to-face classes to online classes, and non-lecture practicums such as anatomy labs and clinical skills training should be implemented in a way that minimizes the risk of infection. To minimize the spread of infection, we made it mandatory for all students and professors to wear masks, keep 2 meters apart, fill out a health condition questionnaire, and measure their body temperature before these classes every day.

In this study, we present our experience in running a medical school curriculum under COVID-19 pandemic conditions by moving all offline classes online and minimizing face-to-face practices. We also present data on student satisfaction, problems, and achievements, and some perspectives on the future.

## Methods

Every course for all years from first-year to fourth-year medical students, except basic laboratory practicums such as anatomy labs and clinical clerkships, was switched to an online program.

### Curricular change

In the first semester of 2020, school opening was postponed due to the regional infections of COVID-19 in February 2020, involving Daegu and Gyeongbuk area. Because of the outbreak, the courses were re-organized and re-opened 2 weeks later online. Online learning was run using the e-Teaching and Learning System, a learning management system of Seoul National University, and was delivered mainly using pre-recorded lecture video clips, with some courses using live online classes.

The first year started with a 1-week integrated medical humanities course followed by basic medical science courses such as anatomy, biochemistry, and physiology. Basic laboratory classes requiring face-to-face contact and which used to run in parallel with the lectures were

postponed until social distancing was loosened in early May. We first moved all basic science lectures online, and for basic laboratory classes students were divided into small groups to reduce the spread of possible infections. To protect our students against infection and toxic material, we provided students with personal protective equipment (PPE) such as face shields and masks. We also asked students to fill out a health condition questionnaire and measured their body temperature before class every day.

The second-year curriculum was mainly composed of an organ-system-based integrated course. When the medical school stopped face-to-face classes in late February, the second-year curriculum was in the middle of the integrated gastrointestinal system course. The first half of the gastrointestinal system course was delivered in the classroom and the other half was delivered online using lecture video clips. The courses that followed, such as those on the respiratory system and circulatory system, were also delivered mainly online using video clips.

The third-year curriculum was composed of the core clinical clerkship covering internal medicine, surgery, etc. As infection rates increased, we put off the clinical clerkship till late April and ran a 2-week online course on integrated medical humanities. We also provided online clinical didactic sessions to allow for later entry into the clinical clerkship. After the social distancing was loosened by the government, students could participate in the core clinical clerkship at the hospital with new guidance under the COVID-19 pandemic situation, and they were not involved in the care of patients with suspected or confirmed COVID-19. Students were required to take preventive measures against the epidemic, such as hand washing and wearing a mask, and were allowed COVID-19 testing if indicated. The fourth-year curriculum consisted of elective clinical clerkships. Similar to the third-year curriculum, online clinical didactic sessions were provided first, and the clerkship was started after the social distancing requirement was loosened. A schematic diagram summarizing the curriculum changes is presented in Fig 1.

## Subjects

The subjects of the study were first-, second-, third-, and fourth-year medical students and professors at Seoul National University College of Medicine (Republic of Korea). In the year 2020, there were 145 students in the first and second years of the medical course, 155 students in the third year, and 149 students in the fourth year. Professors who participated in online courses were included in this study.

**Survey.** After each course, students were asked to complete a questionnaire that included the following items: 1) overall satisfaction with the online course, 2) satisfaction with technical aspects of the online lectures, 3) preference for an online course, 4) strengths of the online course, 5) weaknesses of the online course, and 6) any other comments or suggestions regarding the online course. Students were asked to respond using a 5-point scale that ranged from 1 (very dissatisfied) to 5 (very satisfied). Professors who participated in online courses were asked to complete a questionnaire similar to the students' questionnaire and revised for the professors.

## Academic achievement

While in the midst of the COVID-19 crisis, we had many worries about how to do academic testing. Since proper assessment is a part of learning, and minimizing the spread of infection is also important, there were many concerns about test timing and methods. After each course, we used to evaluate students' academic performance through the test questions made by the professors who ran the course. After the big outbreak, the daily record for new infections remained under 30 cases per day and the requirement for social distancing was relaxed in late

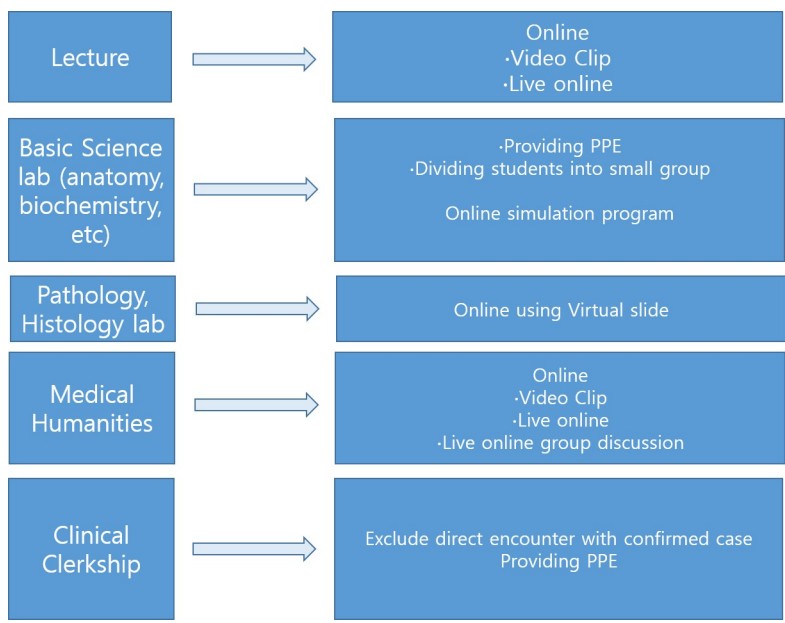

**Fig 1. Schematic diagram of the curriculum.** The courses were re-organized to mainly comprise online program. After the social distancing requirement was loosened by the government, students could participate in face-to-face activities.

April. We decided to proceed with offline examinations because academic misconduct in online examinations is a key concern of many educators [7]. With preventive measures such as hand washing, mask wearing, and keeping 2 meters apart, we divided students into small groups and, accordingly, recruited additional professors and officers to conduct the exams under infection control guidelines. By checking for symptoms such as fever on test day, we ensured that every student with any symptom took the test alone at a prepared place or was instructed to apply for reexamination. The examination was composed of multiple choice questions.

We analyzed the distributions and means of the scores to find out whether there was a difference in students' academic performance with the introduction of online learning.

## Statistical analysis

Statistical analysis was performed using the SPSS (version 23) statistical package (IBM SPSS Statistics) and SAS (version 9.3) statistical package (SAS Institute). We performed the Pearson's chi-squared test as a measure of association to analyze the data. Means were compared using analysis of variance (ANOVA). And we used mixed effects model to identify patterns of score change over years and to determine online class effect on academic achievement. Effect size and 95% confidence intervals were calculated using Cohen's d. $P$ values of $< 0.05$ were taken to indicate statistically significant differences.

## Ethical considerations

The Seoul National University College of Medicine institutional review board provided study approval and waived the requirement for written informed consent (IRB No.2003-159-1111).

## Results

A total 69.7% (318/456) of students and 35.2% (44/125) of professors answered the questionnaire.

## Student satisfaction with the online course

Students were generally satisfied with the online course (Table 1). They answered that "they were generally satisfied with the course (3.97/5)," "the educational objectives of the course were clearly presented (4.14/5)," "the course were organized well (4.08/5)," and "the volume of learning was reasonable (3.85/5)." Among online courses, students were mostly satisfied with the integrated medical humanities course. Comparing the average satisfaction with individual lectures with the previous year, there was no significant difference in overall satisfaction.

As to their satisfaction with technical aspects of online lectures, students answered that they were satisfied with the quality, sound, and speed of the video clips. Students pointed out the following strengths of online learning: 1) they can take the course anywhere they want (4.64/5), 2) they can take the course at any time they want (4.66/5), 3) they can review any portion of the lecture multiple times (4.57/5), 4) they can alter the sequence of the lectures (4.07/5), and 5) they can play the lecture at any speed they want (3.72/5). They pointed out that the weaknesses of online learning are the lack of interaction between the professor and each student and among students. As to difficulty in concentrating during online lectures or difficulty in maintaining self-directed learning, students answered neutrally (2.86/5 and 2.73/5, respectively).

## Faculty satisfaction with the online course

The professors were satisfied with the guide for online lectures (4.05/5), the overall process of online class operation (3.77/5), and the technical aspects of online lectures (3.81/5). They

**Table 1. Students' satisfaction with the online course.**

| Item | Mean (SD) [*] |
|---|---|
| Overall satisfaction on the course | |
| • I am generally satisfied with the course | 3.97 (0.95) |
| • The educational objectives of the course were clearly presented | 4.14 (0.87) |
| • The course lectures were well-organized in relation to each other | 4.08 (0.94) |
| • I am generally satisfied with the volume of learning | 3.85 (1.10) |
| Technical aspects of online lectures | |
| • I am generally satisfied with the progress of the online lecture | 3.96 (1.11) |
| • I am generally satisfied with the video quality of the lecture | 4.13 (0.96) |
| • I am generally satisfied with the sound quality of the lecture | 3.82 (1.17) |
| • I am generally satisfied with the speed of the lecture | 3.92 (1.05) |
| • Feedback via email was done properly | 3.73 (1.06) |
| The strengths of online learning | |
| • Taking the course at any time | 4.64 (0.67) |
| • Taking the course anywhere | 4.66 (0.63) |
| • Flexibility in the sequence of the lecture | 4.07 (1.11) |
| • Playing the lecture at any speed they want | 3.72 (1.33) |
| • Reviewing multiple times any portion of the lecture | 4.57 (0.78) |
| The weaknesses of online learning | |
| • Lack of interaction between professor and student | 3.10 (1.20) |
| • Lack of interaction among students | 3.11 (1.33) |
| • Difficulty in concentrating during online lectures | 2.86 (1.40) |
| • Difficulty in maintaining self-directed learning | 2.73 (1.32) |

5-point Likert scale: 5: strongly agree, 4: agree, 3: neither agree nor disagree, 2: disagree, and 1: strongly disagree.
[*]SD: standard deviation.

pointed out that the strengths of online learning were that "they can give a lecture anywhere (3.68/5) and anytime they want (4.01/5)." As for the weaknesses of online learning, the lack of interaction between professor and student (2.02/5) and difficulty in grasping the students' level of understanding (1.93/5) were suggested (Table 2). They also answered that it took more time and effort to prepare lectures, and that they had difficulty in preparing lecture materials due to copyright issues and personal information protection.

## Preference for online learning

Students' preference for online lectures was much higher than for offline lectures (online vs. offline: 63% vs. 29%), and they preferred recorded video (75.5%) to live online classes (11.3%) in all years and courses (Fig 2). In contrast, professors preferred offline lectures (77.3%) over online lectures (13.6%), and had a higher preference for live online classes (61.3%) than for recorded video (31.9%). As to the survey on future education plans, 84.3% of students wanted to maintain online courses even after the COVID-19 pandemic ends. Among them, 45.5% answered that they wanted to combine offline and online classes, and 38.8% of students answered that they wanted most of the lectures to be maintained online. Although professors provided fewer answers than students who wanted to maintain online lectures, 47.7% of professors also said they hoped to maintain online lectures. Over half of the professors (52.3%) wanted to go back to offline lectures.

**Academic performance.** To compare achievement, we compared examination scores from 2018 to 2020, although the exams were not standardized for difficulty. As there is no significant change in the competencies to be acquired through the course, and the composition of the professors who made test questions are similar, we can expect the difficulty level of the exam would not change much. In some courses, such as the anatomy course and the

**Table 2. Professors' satisfaction with the online course.**

|  | Mean (SD)* |
|---|---|
| Overall satisfaction with the course |  |
| • Guidance on online training was appropriate and easy to understand | 4.05 (0.77) |
| • Online teaching (making a lecture video clip or conducting a live online class) was easy | 3.77 (1.15) |
| • The environment for making the lecture clip was satisfactory | 3.91 (0.95) |
| • There was no inconvenience in booking the place to make the lecture clips | 3.81 (1.06) |
| • I am satisfied with the QnA** process after class | 2.02 (2.02) |
| The strengths of online learning |  |
| • Giving the lecture at any time | 4.02 (0.84) |
| • Giving the lecture anywhere | 3.68 (1.06) |
| • Correcting the part of the lecture flexibly | 3.48 (0.97) |
| • Using the given class time more efficiently | 3.09 (1.04) |
| The weaknesses of online learning |  |
| • Taking more time and effort to prepare for the online lecture | 2.57 (0.86) |
| • Copyright issues make it difficult to prepare lecture materials | 2.66 (0.74) |
| • The computers and related equipment for online lectures are unfamiliar | 3.09 (1.02) |
| • Difficulty in grasping the students' level of understanding | 1.93 (0.94) |
| • Lack of interaction between professor and student | 1.66 (0.93) |

5-point Likert scale: 5: strongly agree, 4: agree, 3: neither agree nor disagree, 2: disagree, and 1: strongly disagree.
*SD: standard deviation.
**QnA: question and answer

### I prefer online class rather than offline class

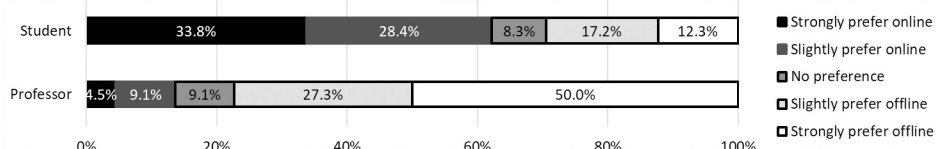

### I prefer recorded video rather than live online class

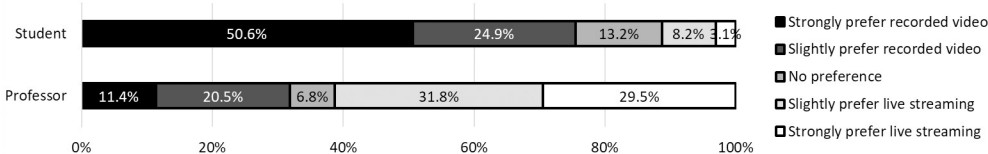

### Preference for future education plans

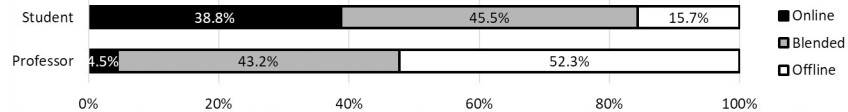

**Fig 2. Students' and professors' preference for online learning.** Students preferred online lectures over offline lectures, and video-recorded lectures over real-time lectures.

respiratory system course, the mean score in 2020 was lower than that in 2018 or 2019 (Table 3). However, since the mean of each subject exam score changes year by year, and the ANOVA analysis only indicates whether the difference in the mean of each year is significant, it is necessary to analyze whether the overall pattern of change is significant. And even if it is significant, it is necessary to analyze how meaningful the amount of change is.

As the exam score is a repeated measurement data that a student participates in several subject tests and is related to each other, a mixed model analysis was performed. For the analysis

**Table 3. Students' examination scores for 3 years (2018–2020).**

|  | 2018 | 2019 | 2020 | *p*-value |
|---|---|---|---|---|
|  | Mean (SD) * | Mean (SD) * | Mean (SD) * |  |
| Anatomy | 86.0 (7.0) | 88.1 (10.3) | 82.0 (11.5) | <0.0001 |
|  | N = 150 | N = 147 | N = 143 |  |
| Biochemistry | 79.7 (11.5) | 70.9(17.1) | 74.1 (17.3) | <0.0001 |
|  | N = 149 | N = 152 | N = 144 |  |
| Histology | 86.2 (6.7) | 85.1 (12.9) | 83.4 (12.0) | 0.0754 |
|  | N = 152 | N = 150 | N = 144 |  |
| Gastrointestinal system | 86.6 (8.8) | 88.4 (10.5) | 85.9 (10.4) | 0.0825 |
|  | N = 150 | N = 153 | N = 145 |  |
| Respiratory system | 78.7 (13.1) | 88.2 (9.2) | 76.9 (11.7) | <0.0001 |
|  | N = 151 | N = 158 | N = 145 |  |
| Circulatory System | 79.2 (10.6) | 80.1 (10.5) | 77.3 (12.1) | 0.0854 |
|  | N = 150 | N = 157 | N = 145 |  |

*SD: standard deviation.

of each subject, subjects were analyzed as fixed effects and the interaction between the two variables (year, subject) was included to check whether there was any difference in the pattern of change by subject. As a result, the p-value was less than 0.0001, indicating that the pattern of change for each subject was significantly different. Therefore, it was analyzed whether there was a difference in scores in 2018, 2019, and 2020 for each subject. At this time, since the online class conversion due to COVID-19 is a big change, we looked at whether there was a difference between the average score in 2018 and 2019 and the score in 2020 in order to investigate this impact.

For anatomy, the average score for 2018 and 2019 was 86.67 and was higher than the average score of 82.55 for 2020. For the rest of the subjects, the scores in 2020 were lower than the average of scores in 2018 and 2019 (the average difference was negative for all subjects). In addition, differences were statistically significant in anatomy, circulatory and respiratory systems (*P*-value = 0.0004, 0.0138, <0.0001) (S1 Table). Using mixed model, we could find out that exam score of some subject showed significant change with the introduction of online class. To analyze the overall score changes over years, the subjects were analyzed as random effects, considering that the degree of difficulty may be different. As a result, there is an overall difference between the years, and the difference between the 2020 score and the average score in 2018 and 2019 was -2.10, which was low in 2020 and was statistically significant (*P* = 0.0001).

Since the ANOVA results and the mixed model results are similar, the effect size of the difference between the average of the 2020 scores and the average of 2018 and 2019 scores was calculated using the mean and standard deviation, not least square mean (Table 4). The effect size of anatomy, respiratory system and circulatory system course score is -0.5150, -0.5504 and -0.2116, respectively. In the case of anatomy and respiratory system course, the change in academic achievement by online class is moderate, and in the case of circulatory system, the change by online class is small.

## Discussion

In this study, we present our experience of moving our classes online and our survey of students and professors for their feedback. We continued offline clinical clerkship, clinical skills training, and basic laboratory classes with preventive measures such as PPE. Contrary to the professors' concerns, students were generally satisfied with the online course and seem to be adjusting very well. Moreover, they preferred online to offline lectures and wanted to maintain the online course even after the COVID-19 pandemic is over. However, professors preferred offline lectures and more than half of them wanted to go back to offline lectures. Students' academic performance did not differ significantly compared with the year before the curricular changes in most courses. In some courses the test scores dropped slightly, but the differences were not significant.

**Table 4. Effect size of students' examination scores of 2020 compared to 2018 and 2019.**

| Subject | Cohen's d effect size |
| --- | --- |
| Anatomy | -0.5150 |
| Biochemistry | -0.0754 |
| Histology | -0.2127 |
| Gastrointestinal system | -0.1605 |
| Respiratory system | -0.5504 |
| Circulatory system | -0.2116 |

The educational effects of online learning have been proven through research. The biggest advantage of online learning is that it is possible to learn at any time and anywhere, using the internet. Online learning also allows for learner-oriented learning. With the introduction of online learning in medical education, each student can study at their own speed and repeat what is needed, ultimately enabling them to learn according to their ability. A systematic review of 59 studies suggested that online learning is equivalent to traditional teaching in terms of knowledge and skills gained, and student satisfaction [8]. In addition, online learning, which uses a variety of multimedia content, can be useful not only in medical classes where photographs, paintings, or other images are used to describe some clinical presentations, but also in the evaluation of the students' academic performance. Based on these advantages, it could be expected that student and professor satisfaction with the courses conducted by online learning in this study would be good. There was a difference in satisfaction depending on the type of course. It is assumed that the reasons for the low level of satisfaction with the basic medicine are that the laboratory classes (cadaver dissection) were not provided in a timely way and total laboratory practice was insufficient due to the relatively short time period allocated compared with 2019. As laboratory practice could enhance students' learning, the low test scores in the anatomy course might have been caused by the shortage of timely anatomy practice sessions.

It is very important to maintain students' academic achievement after the conversion to online class. In our study, moderately decreased exam scores were observed in anatomy and respiratory system courses. It is difficult to make an accurate comparison because the degree of difficulty may vary between tests, but a statistically significant decrease was observed in the above two subjects. In anatomy, the aforementioned lack of practice seems to be the cause, and in respiratory system course, we judged that it was difficult to compare precisely because of the unusually large annual variation in difficulty index. However, apart from these reasons, if the academic achievement of medical students really declines due to online learning, this is a serious problem. It is necessary to observe whether actual academic achievement decreases, and if so, to find out how to resolve this decrease. In some way, it may be predictable that the efficiency of medical education decreases when practice is insufficient.

Interestingly, students preferred recorded lectures over live online lectures, and professors preferred live online lectures. This finding is in contrast to the result of Brockfeld et al. that students preferred live online lectures [9]. Recently, Ashokka et al. introduced the transition of lectures to online streaming with interactive components set according to the pandemic alert level [10–12]. Students in this study were dissatisfied with the disadvantages of live online lectures, such as being in front of laptop in a fixed class and taking the class once without repeating it.

As for the limitations of online learning, the practical problems associated with the design and development of online learning programs are drawbacks. Professors' conservative tendencies and reluctance also serve as obstacles to the spread and long-term adoption of online learning in medical education. As many scholars [13–15] have pointed out, professors familiar with traditional teaching methods are reluctant to introduce online learning to their courses and be burdened with the current situation of having to introduce online learning. Active faculty support at the college level, and close cooperation through multiple meetings between schools, faculty, and students, helped ease this situation [10].

Since the COVID-19 outbreak rapidly transitioned into a worldwide pandemic, we are facing unprecedented times. This pandemic has disrupted medical education and will change many things and make it difficult to go back to the past. The term "new normal" has been coined [16]. Even before the COVID-19 era, online lectures were already showing their effectiveness and were being used by many educational institutions. This pandemic made offline lectures disappear and most lectures are currently delivered online. However, there is still no

substitute for clinical clerkship, which is the core curriculum of medical schools. Virtual clinical learning, virtual care, hospital at home, and other innovations are being proposed as a complement to the clinical clerkship, but the implication is still a relatively limited learning experience [17]. We are maintaining this while devising ways to minimize risk, worrying that students may potentially spread the virus when asymptomatic and may acquire the virus in the course of training. In Korea, during the Daegu outbreak crisis, all medical schools suspended their academic schedules, as did the United States and other countries experiencing their own regional outbreaks. As the crisis slowly passed, the school schedule slowly resumed. Medical schools should make decisions that balance student safety from COVID-19 infection with training students with sufficient clinical experience. Decisions include triaging which activities should be continued, postponed, adapted, dropped, or added [18]. We continued the lectures by putting them online, postponed the clinical clerkship and basic medical practice, dropped a few parts of the clerkship with high risk of infection, and added a regular live online discussion session to help students lead self-directed learning.

Our study has several limitations. First, our study was performed at a single institution. As each medical school has different situations and circumstances, our curricular change and results may not be generalizable to other institutions. Second, as we used exam scores that were not standardized for difficulty level, accurate comparison of academic achievement with the introduction of online class. If we had used nation-wide examination or item response theory based computer adaptive test, a more accurate comparison would have been possible. Finally, our exam was MCQ test that evaluates student's academic achievement focused on cognitive domain. To assess of students' achievement related to psychomotor or affective domain, it would have been necessary to use other assessment tool.

The medical education environment is changing rapidly with COVID-19 and we are only at the beginning. COVID-19 may forever change how future physicians are educated. Further research is needed to maximize the benefits of online education, compensate for any shortcomings and try various educational attempts.

## Supporting information

**S1 Fig. Box and whisker plot the test score.**
(PPTX)

**S1 Table. Students' examination scores in 2020 compared to in 2018 and 2019.**
(DOCX)

**S1 File. Anonymized data set.**
(XLSX)

**S1 Appendix. Questionnaire.**
(DOCX)

## Acknowledgments

The authors acknowledge all students, professors, and teaching assistants in Seoul National University College of Medicine for helping us move our curriculum smoothly online and the Medical Research Collaborating Center at Seoul National University Hospital for their support for statistical analyses.

## Author Contributions

**Conceptualization:** Ju Whi Kim, Sun Jung Myung.

**Data curation:** Hyun Bae Yoon, Sang Hui Moon.

**Formal analysis:** Ju Whi Kim, Sun Jung Myung.

**Investigation:** Ju Whi Kim, Sun Jung Myung.

**Methodology:** Hyun Bae Yoon, Sang Hui Moon, Hyunjin Ryu, Jae-Joon Yim.

**Project administration:** Sun Jung Myung.

**Supervision:** Sun Jung Myung.

**Writing – original draft:** Ju Whi Kim, Sun Jung Myung, Hyun Bae Yoon, Sang Hui Moon, Hyunjin Ryu, Jae-Joon Yim.

**Writing – review & editing:** Ju Whi Kim, Sun Jung Myung, Hyun Bae Yoon, Sang Hui Moon, Hyunjin Ryu, Jae-Joon Yim.

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
