## [Decision Letter · Decision Letter 0]

5 Oct 2020

PONE-D-20-26059

How medical education survives and evolves during COVID-19: our experience and future direction

PLOS ONE

Dear Dr. Myung,

Thank you for submitting your manuscript to PLOS ONE. After careful consideration, we feel that it has merit but does not fully meet PLOS ONE’s publication criteria as it currently stands. Therefore, we invite you to submit a revised version of the manuscript that addresses the points raised during the review process.

The work is of interest and address issues we are all, actually facing. Please, carefully revise  your manuscript accordingly with the expert reviewers comments with a response letter   (point by point). Submit a revised version, at your earliest convenience.

We look forward to receiving your revised manuscript.

Kind regards,

Cesario Bianchi

Academic Editor

PLOS ONE

Journal Requirements:

2.We note that you have indicated that data from this study are available upon request. PLOS only allows data to be available upon request if there are legal or ethical restrictions on sharing data publicly. For information on unacceptable data access restrictions, please see http://journals.plos.org/plosone/s/data-availability#loc-unacceptable-data-access-restrictions.

Additional Editor Comments (if provided):

Dear Dr. Myung:

Thank you for submitting your work. It was reviewed by 2 experts that found the data of interest. Reviewer #1, and myself, found that the manuscript could be more focused and the statistical analyses clearly stated and the in line with the discussion. Please, carefully revise your interesting work accordingly with the reviewers suggestions and , if you find appropriate, incorporate changes in your revised version.

Reviewers' comments:

Reviewer's Responses to Questions

**Comments to the Author**

1. Is the manuscript technically sound, and do the data support the conclusions?

Reviewer #1: Yes

Reviewer #2: Partly

2. Has the statistical analysis been performed appropriately and rigorously? 

Reviewer #1: Yes

Reviewer #2: No

3. Have the authors made all data underlying the findings in their manuscript fully available?

Reviewer #1: Yes

Reviewer #2: No

4. Is the manuscript presented in an intelligible fashion and written in standard English?

Reviewer #1: Yes

Reviewer #2: Yes

5. Review Comments to the Author

Reviewer #1: The theme discussed at the article is very important and currently. It's important to the other Medical Schools to know what is being done about medical education. There are a lot of medical's studentes around the world who have been affected by the pandemic, like other students. But, the Medical Schools have many hours in practical and theoretical classes, and continue the medical education is a challenge.

Reviewer #2: Overall, I think this is a very interesting and timely paper as it captures real life data on the sudden change in curriculum that so many educators have been faced with. Unfortunately, this paper in some ways tries to do too many things. It gives an example of how a cirrculum can be adjusted for the current conditions, but also tries to make statements about satisfaction with the online cirriculum. I think the authors would be better served by focusing on the latter goal, and briefly describing their cirrciculum, then moving into the data driven portion of the manuscript and centering the manuscript more around the actual study than just the logistics of education during COVID.

I have several additional comments/concerns:

- For the evaluation of academic performance, what tests were used? Were these national standardized tests or tests specific to the university. Given that you make important comments about the academic performance not suffering from this change in curriculum, we need more details about the tests that were used to do this evaluation.

- Also, on the topic of testing, you mention that tests were not standardized from year to year, how are these tests created and how much do they change year to year?

- In the statistical analysis of test performance, what was the N for each class? Is the N high enough to assume normal distribution? If not, should this be reported as a median? Also, what is the standard deviation with median or standard error with mean? Need some idea of the distribution.

- There are differences in how much the academic performance changed between the different subjects. Is this worth looking into further or discussing further? Was this a trend between year groups (1st vs 2nd year) or any other trend noted here? Potentially something to look at and/or discuss.

- Finally, you concluded no significant differences in academic performance, yet almost all of the the p values you report are statistically significant, this must be addressed. What is your threshold for “significant” differences, the language here needs to be clear. Also, if there are subtle differences, this should not be ignored as the trend is certainly towards more online learning, so are there ways to improve this and make academic performance better?

- You state that “Students pointed out the following strengths of online learning…” What exactly does this mean? Was there a focus group where students pointed out benefits and problems and then the Leikert scale on survey was used to assess agreement with these statements? Or did the authors come up with the statements used for the survey? The language here could be more clear. If the point is just that the score on the survey of these states was higher than three, the language would be better as “Students agreed that the following were strengths of…” something like that. If this was a focus group coming up with strengths/weaknesses, this should be pointed out.

- The portions of the manuscript describing the specifics of how students were able to go to labs and clinical clerkships are of interest, but I think the analysis of surveys and academic performance are interesting as they give some data on how things went. I would emphasize these areas and tighten up the data as above.

- Authors should discuss the limitations of their study in the discussion.

6. PLOS authors have the option to publish the peer review history of their article (what does this mean?). If published, this will include your full peer review and any attached files.

Reviewer #1: No

Reviewer #2: **Yes: **Timothy Vreeland

---

## [Author Response · Author response to Decision Letter 0]

5 Nov 2020

PLOS ONE/ PONE-D-20-26059

We thank the Reviewer for his/her thoughtful and expert review of our manuscript and for their valuable and insightful comments. We have responded to each of the Reviewer’s comments and have incorporated all modifications suggested by the reviewer into the revised manuscript. The changes within the revised manuscript were highlighted (underlined and in blue). Our responses to the Reviewer’s comments are as follows:

Journal Requirements:

Author’s Response: According to the Reviewer’s comment, we have confirmed our manuscript meets PLOS ONE's style requirements.

2. If there are no restrictions, please upload the minimal anonymized data set necessary to replicate your study findings as either Supporting Information files or to a stable, public repository and provide us with the relevant URLs, DOIs, or accession numbers. Please see http://www.bmj.com/content/340/bmj.c181.long for guidelines on how to de-identify and prepare clinical data for publication. For a list of acceptable repositories, please see http://journals.plos.org/plosone/s/data-availability#loc-recommended-repositories.

Author’s Response: According to the Reviewer’s comment, we have uploaded the minimal anonymized data set necessary to replicate our study findings as Supporting Information files.

Additional Editor Comments

Thank you for submitting your work. It was reviewed by 2 experts that found the data of interest. Reviewer #1, and myself, found that the manuscript could be more focused and the statistical analyses clearly stated and the in line with the discussion. Please, carefully revise your interesting work accordingly with the reviewers suggestions and, if you find appropriate, incorporate changes in your revised version.

Author’s Response: According to the Reviewer’s comment, we have made the manuscript focused and the statistical analyses clearly stated and the in line with the discussion. Please refer to the following answers. 

Review Comments to the Author

Reviewer 1 Comments:

1. Reviewer’s comment: The theme discussed at the article is very important and currently. It's important to the other Medical Schools to know what is being done about medical education. There are a lot of medical's students around the world who have been affected by the pandemic, like other students. But, the Medical Schools have many hours in practical and theoretical classes, and continue the medical education is a challenge.

Author’s Response: 

Thank you for the reviewer's encouragement. As the Reviewer’s comment, a lot of medical schools around the world are struggling to educate follow-up generations of doctors even in COVID-19 pandemic situation. We will find a breakthrough to innovate medical education even in the current chaotic situation through ceaseless efforts and communication.

Reviewer 2 Comments:

1. Reviewer’s comment: Overall, I think this is a very interesting and timely paper as it captures real life data on the sudden change in curriculum that so many educators have been faced with. Unfortunately, this paper in some ways tries to do too many things. It gives an example of how a curriculum can be adjusted for the current conditions, but also tries to make statements about satisfaction with the online curriculum. I think the authors would be better served by focusing on the latter goal, and briefly describing their curriculum, then moving into the data driven portion of the manuscript and centering the manuscript more around the actual study than just the logistics of education during COVID.

Author’s Response: The authors agree with the reviewer’s comment that focusing on the actual study than just the logistics of education during COVID. However, to present students’ satisfaction with the online curriculum and their academic achievement, describing our effort to adjust our curriculum under pandemic situation and how the curriculum is like was an indispensable requirement. As the Reviewer pointed out, we have more focused on the latter portion. According to the Reviewer’s comment, we have added the sentence “As there is no significant change in the competencies to be acquired through the course, and the composition of the professors who made test questions are similar, we can expect the difficulty level of the exam would not change much. In some courses, such as the anatomy course and the respiratory system course, the mean score in 2020 was lower than that in 2018 or 2019 (Table 3). However, since the mean of each subject exam score changes year by year, and the ANOVA analysis only indicates whether the difference in the mean of each year is significant, it is necessary to analyze whether the overall pattern of change is significant. And even if it is significant, it is necessary to analyze how meaningful the amount of change is.” on page 11-12, lines 202-210, and “As the exam score is a repeated measurement data that a student participates in several subject tests and is related to each other, a mixed model analysis was performed. For the analysis of each subject, subjects were analyzed as fixed effects and the interaction between the two variables (year, subject) was included to check whether there was any difference in the pattern of change by subject. As a result, the p-value was less than 0.0001, indicating that the pattern of change for each subject was significantly different. Therefore, it was analyzed whether there was a difference in scores in 2018, 2019, and 2020 for each subject. At this time, since the online class conversion due to COVID-19 is a big change, we looked at whether there was a difference between the average score in 2018 and 2019 and the score in 2020 in order to investigate this impact.

Among subjects in Table 3, anatomy, biochemistry and histology are 1st grade subjects whereas gastrointestinal system, respiratory system and circulatory system are 2nd grade subjects. For anatomy, the average score for 2018 and 2019 was 86.67 and was higher than the average score of 82.55 for 2020. For the rest of the subjects, the scores in 2020 were lower than the average of scores in 2018 and 2019 (the average difference was negative for all subjects). In addition, differences were statistically significant in anatomy, circulatory and respiratory systems (p-value=0.0004, 0.0138, <0.0001) (Supplementary Table 1). Using mixed model, we could find out that exam score of some subject showed significant change with the introduction of online learning.

Since the ANOVA results and the mixed model results are similar, the effect size of the difference between the average of the 2020 scores and the average of the scores in 2018 and 2019 was calculated using the mean and SD, not lsmean (Table 4). The effect size of anatomy, respiratory system and circulatory system is -0.5150, -0.5504 and -0.2116, respectively. In the case of anatomy and respiratory system course, the change in academic achievement by e-learning is moderate, and in the case of circulatory system, the change by e-learning is small.” on page 12-13, lines 215-242 within the revised manuscript.

2. Reviewer’s comment: For the evaluation of academic performance, what tests were used? Were these national standardized tests or tests specific to the university. Given that you make important comments about the academic performance not suffering from this change in curriculum, we need more details about the tests that were used to do this evaluation.

Author’s Response: We evaluated students’ academic performance using tests specific to the university. As national standardized tests are usually scheduled the end of the semester, we couldn’t apply those test to compare students’ academic performance with or without curricular change under COVID-19. According to the Reviewer’s comment, we have added the sentence “After each course, we used to evaluate students’ academic performance through exam that was developed by the professors who ran the course.” on page 6, lines 113-115 and “As there is no significant change in the competencies to be acquired through the course, and the composition of the professors who made test questions are similar, we can expect the difficulty level of the exam would not change much.” on page 11-12, lines 202-205 within the revised manuscript.

3. Reviewer’s comment: Also, on the topic of testing, you mention that tests were not standardized from year to year, how are these tests created and how much do they change year to year?

Author’s Response: According to the Reviewer’s comment, we have added the sentence “After each course, we used to evaluate students’ academic performance through exam that was developed by the professors who ran the course. As there is no significant change in the competencies to be acquired through the course, and the composition of the professors who developed the exam is similar, the exam was developed as done as previous year we can expect the difficulty level of the exam would not change much.” on page 6, lines 114-119 within the revised manuscript and also refer the answer.

4. Reviewer’s comment: In the statistical analysis of test performance, what was the N for each class? Is the N high enough to assume normal distribution? If not, should this be reported as a median? Also, what is the standard deviation with median or standard error with mean? Need some idea of the distribution.

Author’s Response: According to the Reviewer’s comment, we have added the N for each class in Table 3 on page 12, lines 212-213 within the revised manuscript. The N for each data is from 143 to 158, which is high enough to assume normal distribution. We also added supplementary figure which present the distribution of the test score. (Supplementary Figure 1) The meaning of mean score and standard deviation in our study describe the distribution of the test scores of students. 

5. Reviewer’s comment: There are differences in how much the academic performance changed between the different subjects. Is this worth looking into further or discussing further? Was this a trend between year groups (1st vs 2nd year) or any other trend noted here? Potentially something to look at and/or discuss.

Author’s Response: Among subjects in Table 3, anatomy, biochemistry and histology are 1st grade subjects whereas gastrointestinal system, respiratory system and circulatory system are 2nd grade subjects. There is no significant change in average score that could be attributed to difference of year groups (1st vs 2nd year). As the exam score is a repeated measurement data that a student participates in several subject tests and is related to each other, a mixed model analysis was performed. 

6. Reviewer’s comment: Finally, you concluded no significant differences in academic performance, yet almost all of the p values you report are statistically significant, this must be addressed. What is your threshold for “significant” differences, the language here needs to be clear. Also, if there are subtle differences, this should not be ignored as the trend is certainly towards more online learning, so are there ways to improve this and make academic performance better?

Author’s Response: We reviewed the data and consulted statistician (Professor Choi). Then, we came to have final results which presented in result part within the revised manuscript. 

According to the Reviewer’s comment, we have added the sentence “Statistical analysis was performed using the SPSS (version 23) statistical package (IBM SPSS Statistics) and SAS (version 9.3) statistical package (SAS Institute). We performed the Pearson’s chi-squared test as a measure of association to analyze the data. Means were compared using analysis of variance (ANOVA). And we used mixed effects model to identify patterns of score change over years and to determine e-learning effect on academic achievement. Effect size and 95% confidence intervals were calculated using Cohen’s d. P values of <0.05 were taken to indicate statistically significant differences.” on page 7, lines 128-134, “As there is no significant change in the competencies to be acquired through the course, and the composition of the professors who made test questions are similar, we can expect the difficulty level of the exam would not change much. In some courses, such as the anatomy course and the respiratory system course, the mean score in 2020 was lower than that in 2018 or 2019 (Table 3). However, since the mean of each subject exam score changes year by year, and the ANOVA analysis only indicates whether the difference in the mean of each year is significant, it is necessary to analyze whether the overall pattern of change is significant. And even if it is significant, it is necessary to analyze how meaningful the amount of change is. 

As the exam score is a repeated measurement data that a student participates in several subject tests and is related to each other, a mixed model analysis was performed. For the analysis of each subject, subjects were analyzed as fixed effects and the interaction between the two variables (year, subject) was included to check whether there was any difference in the pattern of change by subject. As a result, the p-value was less than 0.0001, indicating that the pattern of change for each subject was significantly different. Therefore, it was analyzed whether there was a difference in scores in 2018, 2019, and 2020 for each subject. At this time, since the online class conversion due to COVID-19 is a big change, we looked at whether there was a difference between the average score in 2018 and 2019 and the score in 2020 in order to investigate this impact.

For anatomy, the average score for 2018 and 2019 was 86.67 and was higher than the average score of 82.55 for 2020. For the rest of the subjects, the scores in 2020 were lower than the average of scores in 2018 and 2019 (the average difference was negative for all subjects). In addition, differences were statistically significant in anatomy, circulatory and respiratory systems (p-value=0.0004, 0.0138, <0.0001) (Supplementary Table 1). Using mixed model, we could find out that exam score of some subject showed significant change with the introduction of online class. To analyze the overall score changes over years, the subjects were analyzed as random effects, considering that the degree of difficulty may be different. As a result, there is an overall difference between the years, and the difference between the 2020 score and the average score in 2018 and 2019 was -2.10, which was low in 2020 and was statistically significant (p=0.0001).

Since the ANOVA results and the mixed model results are similar, the effect size of the difference between the average of the 2020 scores and the average of 2018 and 2019 scores was calculated using the mean and standard deviation, not lease square mean (Table 4). The effect size of anatomy, respiratory system and circulatory system course score is -0.5150, -0.5504 and -0.2116, respectively. In the case of anatomy and respiratory system course, the change in academic achievement by online class is moderate, and in the case of circulatory system, the change by online class is small.” on page 11-14, lines 202-245 and “It is very important to maintain students’ academic achievement after the conversion to online class. In our study, moderately decreased exam scores were observed in anatomy and respiratory system courses. It is difficult to make an accurate comparison because the degree of difficulty may vary between tests, but a statistically significant decrease was observed in the above two subjects. In anatomy, the aforementioned lack of practice seems to be the cause, and in respiratory system course, we judged that it was difficult to compare precisely because of the unusually large annual variation in difficulty index. However, apart from these reasons, if the academic achievement of medical students really declines due to online learning, this is a serious problem. It is necessary to observe whether actual academic achievement decreases, and if so, to find out how to resolve this decrease. In some way, it may be predictable that the efficiency of medical education decreases when practice is insufficient.” on page 15, lines 276-286 within the revised manuscript.

7. Reviewer’s comment: You state that “Students pointed out the following strengths of online learning…” What exactly does this mean? Was there a focus group where students pointed out benefits and problems and then the Leikert scale on survey was used to assess agreement with these statements? Or did the authors come up with the statements used for the survey? The language here could be more clear. If the point is just that the score on the survey of these states was higher than three, the language would be better as “Students agreed that the following were strengths of…” something like that. If this was a focus group coming up with strengths/weaknesses, this should be pointed out.

Author’s Response: In the survey, we included question “Please indicate a 5-point scale to see which of the online lectures you think you are satisfied with compared to the existing offline lectures.” and also asked students to check Likert scale on following question; 1) they can take the course anywhere they want, 2) they can take the course at any time they want, 3) they can review any portion of the lecture multiple times, 4) they can alter the sequence of the lectures, and 5) they can play the lecture at any speed they want. We also attached survey form (Appendix) within the revised manuscript.

8. Reviewer’s comment: The portions of the manuscript describing the specifics of how students were able to go to labs and clinical clerkships are of interest, but I think the analysis of surveys and academic performance are interesting as they give some data on how things went. I would emphasize these areas and tighten up the data as above.

Author’s Response: According to the Reviewer’s comment, we have added further data analysis results. These sentences are described in author's response to reviewer's comment 6. 

9. Reviewer’s comment: Authors should discuss the limitations of their study in the discussion.

Author’s Response: According to the Reviewer’s comment, we have added the sentence “Our study has several limitations. First, our study was performed at a single institution. As each medical school has different situations and circumstances, our curricular change and results may not be generalizable to other institutions. Second, as we used exam scores that were not standardized for difficulty level, accurate comparison of academic achievement with the introduction of online class. If we had used nation-wide examination or item response theory based computer adaptive test, a more accurate comparison would have been possible. Finally, our exam was MCQ test that evaluates student’s academic achievement focused on cognitive domain. To assess of students’ achievement related to psychomotor or affective domain, it would have been necessary to use other assessment tool.” on page 16, lines 322-330 within the revised manuscript.

---

## [Decision Letter · Decision Letter 1]

2 Dec 2020

How medical education survives and evolves during COVID-19: our experience and future direction

PONE-D-20-26059R1

Dear Dr. Myung,

We’re pleased to inform you that your manuscript has been judged scientifically suitable for publication and will be formally accepted for publication once it meets all outstanding technical requirements.

Kind regards,

Cesario Bianchi

Academic Editor

PLOS ONE

Additional Editor Comments (optional):

Dear Myung:

Thank you for carefully revise your manuscript in line with the reviewer comments. I find your manuscript acceptable for publication. Congratulations.

Reviewers' comments:

Reviewer's Responses to Questions

**Comments to the Author**

1. If the authors have adequately addressed your comments raised in a previous round of review and you feel that this manuscript is now acceptable for publication, you may indicate that here to bypass the “Comments to the Author” section, enter your conflict of interest statement in the “Confidential to Editor” section, and submit your "Accept" recommendation.

Reviewer #2: All comments have been addressed

2. Is the manuscript technically sound, and do the data support the conclusions?

Reviewer #2: Yes

3. Has the statistical analysis been performed appropriately and rigorously? 

Reviewer #2: Yes

4. Have the authors made all data underlying the findings in their manuscript fully available?

Reviewer #2: Yes

5. Is the manuscript presented in an intelligible fashion and written in standard English?

Reviewer #2: Yes

6. Review Comments to the Author

Reviewer #2: All comments have been addressed. The authors have done a thorough job of examining and explaining their data.

7. PLOS authors have the option to publish the peer review history of their article (what does this mean?). If published, this will include your full peer review and any attached files.

Reviewer #2: **Yes: **Timothy J Vreeland, MD

---

## [Editor Report · Acceptance letter]

11 Dec 2020

PONE-D-20-26059R1 

How medical education survives and evolves during COVID-19: our experience and future direction 

Dear Dr. Myung:

I'm pleased to inform you that your manuscript has been deemed suitable for publication in PLOS ONE. Congratulations! Your manuscript is now with our production department. 

Kind regards, 

on behalf of

Dr. Cesario Bianchi 

Academic Editor

PLOS ONE